Seventeen-year study reveals fluctuations in key ecological indicators on two reef crests in Cuba

Ramos Amanda 1 2
González-Díaz Patricia 2
Banaszak Anastazia T. banaszak@cmarl.unam.mx 3
Perera Orlando 2
Hernandez Delgado Fredy 2
Delfín de León Sandra 2
Vicente Castro Patricia 2
Aguilera Pérez Gabriela Caridad 2
Duran Alain 4
1 Posgrado en Ciencias del Mar y Limnología, Universidad Nacional Autónoma de México , Mexico City , México
2 Centro de Investigaciones Marinas, Universidad de La Habana , La Habana , Cuba
3 Unidad Académica de Sistemas Arrecifales, Universidad Nacional Autónoma de México , Puerto Morelos , Mexico
4 Department of Biological Sciences, Florida International University , Miami , FL , United States of America
Costantini Federica
Electronic publication date: 2024 Jan 23
Publication date: 2024
Volume: 12
Electronic Location ID: e16705
Received 2023 Jun 6; Accepted 2023 Nov 30
Copyright: ©2024 Ramos et al.
Copyright year: 2024
Copyright holder: Ramos et al.
License: This is an open access article distributed under the terms of the Creative Commons Attribution License, which permits unrestricted use, distribution, reproduction and adaptation in any medium and for any purpose provided that it is properly attributed. For attribution, the original author(s), title, publication source (PeerJ) and either DOI or URL of the article must be cited.
License URL: https://creativecommons.org/licenses/by/4.0/

Keywords: Acropora palmata, Diadema antillarum, Coral cover, Coral health, Coral density

Funding: The Harte Research Institute at Texas A&M University-Corpus Christi CONACYT (Consejo Nacional de Ciencia y Tecnología, México) This work was supported by The Harte Research Institute at Texas A&M University-Corpus Christi. Amanda Ramos Romero was supported by a scholarship from CONACYT (Consejo Nacional de Ciencia y Tecnología, México). The funders had no role in study design, data collection and analysis, decision to publish, or preparation of the manuscript.

==============================
Reef crests in the Caribbean have lost approximately 80% of the foundational habitat-forming coral Acropora palmata (Lamarck, 1816), with declines registered as early as the 1950s mainly from anthropogenic causes. We studied two reef crests in the northwestern region of Cuba over 17 years (2005 to 2021) to evaluate temporal changes in coral cover, dominated by A. palmata, and their potential drivers. The density of A. palmata generally showed a negative trend at both reefs, with the lowest density recorded in 2021 at 0.2 ± 0.05 col. m−2 at Playa Baracoa and 1.0 ± 0.1 col. m−2 at Rincon de Guanabo. The mean size of the colonies in the two reefs also decreased over time. In Playa Baracoa, the mean diameter of A. palmata colonies decreased from 2012 at 67 ± 5.9 cm to 2013 at 34 ± 2.2 cm, whereas in Rincon de Guanabo, a change in diameter was evident from 2015 at 44.3 ± 2.3 to 2021 at 21.6 ± 0.9 cm. Adult colonies (10 cm–50 cm diameter) predominated in most years on both reefs. The populations of A. palmata on both reefs were healthy, with an average of 70% colonies in good condition during the study period. However, A. palmata cover decreased by almost half by 2021, to 8.6% in Playa Baracoa and 16.8% in Rincon de Guanabo. By contrast, macroalgal cover increased two-fold to 87.1% in Playa Baracoa and four-fold to 77.2% in Rincon de Guanabo. The density of the sea urchin Diadema antillarum was higher in Playa Baracoa than in Rincon de Guanabo. The highest densities were 2.8 ± 0.2 ind. m−2 in Playa Baracoa in 2005 and 0.1 ± 0.03 ind. m−2 in Rincon de Guanabo in 2008. Although our results show an overall decline of A. palmata (density and percent cover) and an increase in macroalgae, these two reef crests are in better condition than most reefs in the Caribbean in terms of the density and health of A. palmata populations, and the density of D. antillarum at Playa Baracoa. Our results are important in establishing a management plan to ensure the condition of these reef crests does not degrade further.

Introduction

For millions of years, coral reefs have thrived under the environmental constraints of rather specific temperature, light, salinity, nutrient, and aragonite saturation requirements (Kleypas, McManus & Menez, 1999; Cyronak et al., 2020; Guan et al., 2020). Changes to one or more of these factors can alter ecological processes, community composition, ecosystem functioning, and reef resilience (González-Barrios & Álvarez-Filip, 2018; Estrada-Saldívar et al., 2019; Morais et al., 2020; Setter, Franklin & Mora, 2022). Coral reef degradation over the last decades has been attributed to multiple natural and anthropogenic stressors acting at global, regional, and local scales (Jackson et al., 2014; Bruno & Valdivia, 2016; Dutra et al., 2021). The global rise of CO2 concentrations has led to increased sea surface temperature and subsequent mass bleaching events on coral reefs (Hoegh-Guldberg et al., 2007). Regionally, more intense and frequent hurricanes have formed that directly affect coral reef dwelling organisms (Hoegh-Guldberg et al., 2007; Bender et al., 2010; Heron et al., 2016). Locally, degradation has been caused by the discharge of terrestrially-derived nutrients (Roth et al., 2021), increased fishing pressure on both herbivores and carnivores (Pandolfi et al., 2005; Gurney et al., 2013; Shantz, Ladd & Burkepile, 2020), and the collapse of populations of ecologically important species such as Diadema antillarum (Philippi, 1845) (Hughes, 1994). Eutrophication and decreased densities of herbivorous species promote macroalgal abundance, which can alter community composition and, eventually, the resilience capacity of reefs (Mumby, Hastings & Edwards, 2007; Cramer et al., 2017). However, coral reef resilience is highly variable depending on the dominant species of corals and the combination of stressors affecting the system (Lester et al., 2020).

Coral cover has declined by 50% to 80% since 1970 in the Caribbean (Gardner et al., 2003; Jackson et al., 2014). Reef crests have lost approximately 80% of the key habitat-forming coral species (Gladfelter, 1982; Bythell, Pantos & Richardson, 2004) Acropora palmata (Lamarck, 1816) and A. cervicornis (Lamarck, 1816) in the Caribbean (Graus & Macintyre, 1989; Cramer et al., 2020). These species’ three-dimensional, complex morphology provides reef structure, shelter, and feeding areas for reef-dwelling organisms, thus enhancing reef crest diversity (Cramer et al., 2021). Acropora populations began to suffer declines as early as the 1950s from anthropogenic causes (Cramer et al., 2020), and most Caribbean reef crests are dominated by weedy coral species such as Agaricia agaricites (Linnaeus, 1758) and Porites astreoides (Lamarck, 1816) (Aronson & Precht, 2001).

In Cuba, A. palmata populations decreased between 1987 and 1992 (Alcolado, 2008). Recent assessments have indicated that some reef crests are healthy, such as those located in the southwestern (i.e., Faro Cazones; 22.1043°N, −81.5159°W), central (i.e., Cayo Fragoso; 22.7203°N, −79.3631°W) (Caballero-Aragón et al., 2020), and southern (i.e., La Puntica in Jardines de la Reina) (Hernández-Fernández, López & Sotolongo, 2016) parts of the country. However, most reef crests are characterized by a high prevalence of old mortality (>50%) (Hernández-Fernández, López & Sotolongo. 2016; Hernández-Fernández et al., 2019; Caballero-Aragón et al., 2020) and suffer from natural and anthropogenic stressors (González-Díaz et al., 2018; Caballero Aragón et al., 2019).

In the northwestern region of Cuba close to the city of Havana, reefs are considered to be degraded as evidenced by low coral abundance (∼10% cover) and high macroalgal cover (∼65%) that worsens with proximity to the city, probably as a result of the influence of local stressors (Duran et al., 2018; González-Díaz et al., 2018). The main anthropogenic activities that impact the reefs in front of the city are pollutants such as heavy metals and fertilizers (Armenteros et al., 2009; Graham et al., 2011) via drainage from streets and small rivers (González-Díaz, De la Guardia & Gonzalez-Sanson, 2003). In addition, these reefs are characterized by low herbivorous fish biomass (∼12 g m−2) due to high subsistence fishing pressure and very low herbivore abundance of other reef organisms such as Diadema (Duran et al., 2018).

There are two reef crests on opposite sides of the Havana coast: to the west, Playa Baracoa, and to the east, Rincon de Guanabo. These two sites have marked differences in A. palmata, A. agaricites, P. astreoides, and Diadema populations (Caballero, Alcolado & Semidey, 2009; Perera-Pérez et al., 2012; Hernández-Delgado, González-Díaz & Ferrer Rodriguez, 2017; Hernández-Delgado, González-Díaz & Ferrer-Rodríguez, 2018). These characteristics make an excellent study case to elucidate whether key ecological indicators have changed on these reef crests. To this end, we analyzed benthic data collected over 17 years from 2005 to 2021 to evaluate whether the reef crest communities at Playa Baracoa and Rincon de Guanabo have changed significantly and if so, to what extent and in which direction over time.

Materials & methods

Study area

The study was carried out in two shallow (1–3 m) reef crests located in the northwestern region of Cuba (Fig. 1), Playa Baracoa (PB, 23°03′20′N, 82°33′10′W) and Rincon de Guanabo (RG, 23°10′23.63′N, 82°05′57.46′W). The distance between the two reefs is 46.8 km. The reef crest in Playa Baracoa is 764 m long, between 20 and 60 m wide, and located approximately 230 m from Baracoa, a small fishing village. The study site is 2 km east of Santa Ana River, where untreated wastewater from a local educational institution (Latin American School of Medicine, with an average enrollment of 10,000 students) is released. There is no information on pollution rate, type, or impact, but anecdotal knowledge and personal experience indicate that an unpleasant smell emanates from the waters of the Santa Ana River. On the other hand, Rincon de Guanabo is a marine protected area (Protected Natural Landscape/seascape similar to category V IUCN) located approximately 800 m from the coastline, with an extension of approximately 950 m. The reef crest is nearly 3 km east of an oil drilling and extraction area (Boca de Jaruco thermoelectric power station), but data on nutrient load or pollutants (e.g., hydrocarbons) are either absent or unavailable. Both reef crests are influenced by the pressure of subsistence overfishing and there is no effective management plan for these areas (Armenteros, 2000; Castellanos, Lopeztegui & de la Guardia, 2004).

Figure 1 Location of the study areas.

(A) A map of Cuba showing the approximate location of Playa Baracoa (B) and Rincon de Guanabo (C). Also shown are the six study sites (E1-E6 from left to right) on each reef crest. Map data ©2023 Google Earth, Image Landsat/Copernicus, Data SIO, NOAA, U.S. Navy, NGA, GEBCO; Image ©2023 Airbus; Image ©2023 Maxar Technologies.

The climate of this region is characterized by a rainy season (May to October) and a dry season with frequent cold fronts from November to April (Justiz-Águila & González-Pedroso, 2018). The hurricane season extends from June 1st to November 30th (Coll-Hidalgo & Pérez-Alarcón, 2021). Trade winds dominate the region where the prevalent water current is due east with a countercurrent closer to the coast to the west (Mitrani-Arenal & Cabrales-Infante, 2020).

Data collection

From 2005 to 2021, we surveyed the populations of elkhorn coral (Acropora palmata) and reef health in both crests. In 2005, four stations were evaluated in Playa Baracoa, which were established randomly to evaluate the natural variability of the reef crest. For subsequent surveys, a total of six stations separated approximately 100 to 150 m apart were studied at both reefs. We recorded A. palmata density, re-sheeting, colony diameter, health status, benthic composition, and Diadema antillarum density. Logistics-related limitations prevented us from measuring all parameters in all campaigns (See Table S1 for details). For example, in Rincon de Guanabo for 2005, 2012 and 2013 A. palmata density and diameter were not evaluated. In addition, from 2015 to 2021, the temperature was recorded by a sensor (HOBO) in reefs at Playa Baracoa and Rincon de Guanabo (Fig. S1).

The density of A. palmata colonies was determined using a modified visual linear transect as defined by Loya (1972). We counted the colonies along a band transect that was 10 m long by 1 m wide and noted when colonies showed signs of re-sheeting, where live tissue regrows over the skeleton (Jordan-Dahlgren, 1992). A. palmata density (col. m−2) was calculated as the number of corals found within the transect divided by the area of the sampling unit (10 m2). We estimated the sample size or the number of colonies necessary to obtain a 10% precision using the formula proposed by Zar (1996). The level of accuracy was set as a tradeoff between the rigor of the investigation and the logistical possibilities of carrying out the fieldwork. (1) n=1.962∗S2/M∗0.102

where:

S: standard deviation

M: mean of the pilot sample

We estimated A. palmata colony size by measuring the colony’s largest diameter (dm) and height (h). A. palmata diameter was measured considering the most distal branches. For height, the highest branch of the colony was identified, and the vertical distance from this branch to the substrate was measured. Our results analyzed only the diameter because both metrics presented an asymptotic character, i.e., they were correlated (PB: R = 0.75 and RG: R = 0.72; p < 0.001). We used the following size classes: recruits (≤5 cm), juveniles (>5–≤ 10 cm), and adults (>10 cm) based on Ruiz-Zárate and Arias-González (2004), Smith et al. (2005), and Moulding (2005). Size ranges were established considering an average growth rate of around 10 cm year−1 (Bak, Nieuwland & Meesters, 2009). To understand the population structure better, two size ranges (>10–≤ 50 cm, >50 cm) were added and the size distribution was analyzed with greater detail for both reefs in 2021.

The health status of each A. palmata colony was evaluated based on the following indicators: prevalence of diseases, bleaching (the coral polyps were alive, but devoid of symbionts), old mortality (parts of the colony are dead and are colonized by other organisms), recent mortality (parts of the colony are dead, but other organisms have not yet colonized the skeleton), and the presence of bioeroding organisms such as polychaetes and sponges. The prevalence of each indicator in the two A. palmata populations was quantified as the proportion of unhealthy colonies divided by the total number of colonies.

Within the same 10 m × 1 m transects, we also quantified the abundance of coral, macroalgae, and the density of the sea urchin D. antillarum. The abundances of macroalgae and coral were calculated using the line transect as the proportion, i.e., total distance covered by each category divided by the transect length. In this study, for estimating coral cover, we considered all coral species present i.e., Porites spp., Agaricia agaricites, and A. palmata. In 2021, the category of macroalgae was divided into morphofunctional groups: filamentous algae, TAS (turf/algal sediment mats), encrusting red algae, turf algae, articulated calcareous algae, fleshy macroalgae and Dictyota. The latter was separated from the fleshy group because it was much more abundant. The density of sea urchins (ind. m−2) was estimated in the transect area (10 m2).

Data analysis

Descriptive and inferential analyses were performed in the R program (R Core Team, 2016, version 4.0.5). We tested for homogeneity of variance and normality using Levene’s and Shapiro–Wilk’s tests, respectively (R package nortest). When the data did not fit a normal distribution, we performed a square root transformation, if the premises were met, parametric tests were performed. If the data did not fit a normal distribution, we analyzed which statistical distribution they belonged to. We used generalized linear models to evaluate the temporal changes in A. palmata density, coral size and Diadema density. Benthic cover and differences between the two reefs were tested by Mann–Whitney U-tests as the data were not normally distributed. The effect of D. antillarum on macroalgal abundance was tested with a generalized linear model (beta regression) (R package betareg). For further information on the models and variables tested see Table S2. When differences between years were significant, we used Dunn’s test of multiple comparisons using rank sums (non-parametric) and the package PMCMR, DescTools and Tukey test (parametric). Unless otherwise noted, data are presented as mean ± standard deviation (SD).

Results

The density of A. palmata generally showed a negative trend at both reefs between 2005 and 2021. In the Playa Baracoa reef crest, the density of A. palmata showed temporal fluctuation from 2005, with a significant decline between 2008 and 2012 (Fig. 2). The mean colony density of A. palmata for Playa Baracoa in 2005 was 1.8 ± 0.2 col. m−2, which decreased significantly (p < 0.001) by 2012 to 0.4 ± 0.06 col. m−2; an almost five-fold decrease. In 2013, colony density increased significantly with respect to 2012 (p = 0.01) and was maintained through to 2017. However, by 2021, the density of A. palmata decreased significantly (p < 0.001) relative to 2017 to 0.2 ± 0.05 col. m−2. The density of A. palmata in the Rincon de Guanabo reef crest remained similar, with no significant difference found between 2008 and 2017, with a mean of 2.3 ± 0.2 col. m−2. By 2021 colony density had dropped significantly (p < 0.001) to almost half (1.03 ± 0.1 col. m−2) relative to 2017 (Fig. 2). These reefs showed significant differences between them (W = 16180, p < 0.001), when data in common for both reefs were available: 2008, 2015, 2017, and 2021.

Figure 2 Acropora palmata density.

Mean (±SE) density of A. palmata (col. m−2) for each site surveyed (PB: Playa Baracoa and RG: Rincon de Guanabo) during the sampling periods from 2005 to 2021. The letters above each bar indicate when a significant difference was found between years.

In Playa Baracoa, the density of colonies with signs of re-sheeting in 2012 was 0.1 ± 0.3 col. m−2, increasing to 0.3 ± 0.4 col. m−2 by 2013. However, the trend from 2015 to 2017 was to decrease two-fold (0.2 ± 0.4 col. m−2), and by 2021 the level of re-sheeting had dropped to 0.1 ± 0.3 col. m−2. By contrast, the density of re-sheeting in Rincon de Guanabo increased almost two-fold from 2015 (0.2 ± 0.3 col. m−2) to 2017 (0.3 ± 0.5 col. m−2), and by 2021, it reached the highest value (0.7 ± 0.9 col. m−2) during the study for both reef crests.

The mean diameter of the A. palmata colonies in Playa Baracoa decreased over time, while in Rincon de Guanabo, the smallest diameter was registered in 2021 (PB and RG: p < 0.001). In Playa Baracoa, the mean diameter decreased two-fold (p = 0.0002) between 2012 (67 ± 5.9 cm) and 2013 (34 ± 2.2 cm). In Rincon de Guanabo, a change in diameter was evident from 2015 to 2021. The mean diameter decreased significantly (p < 0.001) from 44.3 ± 2.3 cm in 2015 to 21.6 ± 0.9 cm in 2021 (Fig. 3). The size distribution of A. palmata colony diameters in 2021 showed an asymmetric pattern with a bias towards colonies less than 25 cm diameter in both reefs (Fig. 4).

Figure 3 Acropora palmata colony diameters.

Mean (±SE) diameter of A. palmata colonies (cm) for each site surveyed (PB: Playa Baracoa and RG: Rincon de Guanabo) during the sampling periods from 2005 to 2021. The letters above each bar indicate when a significant difference was found between years.

Figure 4 Acropora palmata size distribution.

Distribution of A. palmata size in 2021 for the reefs at Playa Baracoa (green) and Rincon de Guanabo (blue).

In general, in Playa Baracoa, over time, the relative abundance of colonies from 10 to 50 cm diameter increased, whereas it decreased for those ≥ 50 cm diameter. In Rincon de Guanabo there was no change in the size classes until 2021, when adult abundance (>10 cm diameter) decreased and recruit abundance increased (≤5 cm diameter). In Playa Baracoa, the relative abundance of recruits decreased 2.6-fold from 2005 to 2006 and increased almost two-fold by 2012. By 2015, the relative abundance of recruits was the highest in the study period, at 12.6%, whereas in Rincon de Guanabo, the abundance of recruits increased 1.9-fold from 2015 to 2021. The abundance of juveniles remained practically unchanged during the study on both reefs, with no more than 20% of colonies classed in the size range between 5 and 10 cm in diameter. Colonies between 10 and 50 cm in diameter predominated in most years, making up over 50% of the population. On the other hand, the abundance of colonies greater than 50 cm decreased in both reefs, with less than 15% of colonies reaching this size by 2021 (Fig. 5).

Figure 5 Relative abundance of Acropora palmata life stages.

Relative abundance of A. palmata recruits (≤ five cm diameter), juveniles (5–10 cm diameter), subadults (10–50 cm diameter) and adults (>50 cm diameter) in Playa Baracoa (PB) and Rincon de Guanabo (RG) during the sampling period from 2005 to 2021.

In both crests, most colonies were healthy, averaging 70% during the study period. The highest abundance of healthy colonies was found in 2012 (96%) in Playa Baracoa and 2006 (85%) in Rincon de Guanabo. By contrast, a lower abundance of healthy colonies was recorded in 2016 (PB: 54%, RG: 57%) and 2017 (PB: 59%, RG: 66%) in both reefs. The remaining colonies presented old mortality and bioeroding organisms. In Playa Baracoa, between 2005 to 2013, old mortality affected approximately 32% of the colonies. From 2015 to 2021, bioeroding organisms such as sponges and polychaetes dominated, affecting about 20% of the colonies. In Rincon de Guanabo, for 2006, old mortality predominated, affecting 11% of the population, whereas bioerosion dominated for the rest of the years in 14% of the colonies. The prevalence of bleaching showed a similar tendency in both reefs during the study period. In 2015, bleaching increased, affecting 4% and 8% of colonies in Playa Baracoa and Rincon de Guanabo, respectively, and in 2016, it increased to 18% in both populations of A. palmata. The percentage of bleached colonies decreased from 2017 to 2021 in both reefs, with 2% for the last study year. The highest prevalence of recent mortality (11% of colonies) occurred in Playa Baracoa by 2012, decreasing to 3% by 2016. In both crests in 2017, recent mortality increased to 6% in Playa Baracoa and 5% in Rincon de Guanabo. In 2021, the number of colonies affected decreased 12-fold in Playa Baracoa (0.5%) and 6.3-fold in Rincon de Guanabo (0.8%) relative to 2017. Diseases such as white pox or white band were recorded only in Playa Baracoa, affecting less than 1% of the colonies (Fig. 6).

Figure 6 Prevalence of Acropora palmata affectations.

Prevalence of unhealthy colonies of A. palmata in (A) Playa Baracoa and (B) Rincon de Guanabo during the sampling periods between 2005 and 2021, due to the presence of bioeroding organisms (BIO), signs of bleaching (Bl), old mortality (OM) or recent mortality (RM).

Coral cover decreased significantly between 2015 and 2021 in both Playa Baracoa (from 14.9% to 8.6%; W = 2740.5, p < 0.001) and Rincon de Guanabo (from 27.5% to 16.8%; W = 2737, p < 0.001). By contrast, macroalgal cover significantly increased from 2015 to 2021 in Playa Baracoa, almost doubling from 44.5% to 87.1% (W = 43.5, p < 0.001). Macroalgal cover increased four-fold in Rincon de Guanabo from 22.3% to 77.2% (W = 47.5, p < 0.001) (Fig. 7). In 2021, the benthic substrate in Playa Baracoa was characterized by 20% fleshy macroalgae, 17% TAS (turf/algal sediment mats), 15% encrusting red algae, 14% turf algae, 8% articulated calcareous algae, 7% Dictyota and 3% filamentous algae. In the reef at Rincon de Guanabo, there was a higher cover of 34% encrusting red algae, 17% Dictyota, 9% TAS, 8% turf algae, 7% filamentous algae, 6% articulated calcareous algae, and the lowest cover was of fleshy macroalgae with 4%. Together with an increase in algae, coral cover has been decreasing significantly (PB and RG: p < 0.001).

Figure 7 Macroalgal and coral cover.

Mean (±SE) macroalgal (dark gray) and coral (light gray) cover at each site surveyed (PB: Playa Baracoa and RG: Rincon de Guanabo) in the first sampling period in 2015 and the final one in 2021.

The density of D. antillarum were significantly greater at Playa Baracoa (1.7 ± 1.1 ind. m−2) than at Rincon de Guanabo (0.1 ± 0.2 ind. m−2; p < 0.001). Diadema densities were consistently low at Rincon de Guanabo over the study period (range: 0.03 ± 0.01 ind. m−2 to 0.1 ± 0.03 ind. m−2), whereas at Playa Baracoa there was a decrease from the highest value in 2005, at 2.8 ± 0.2 ind. m−2 to 1.7 ±0.1 ind. m−2 in 2008, after which no significant variations in Diadema densities were recorded. In 2021, densities were 1.9 ± 0.2 ind. m−2 at Playa Baracoa (Fig. 8). No significant effect (p = 0.5) of D. antillarum on algae cover was found in Playa Baracoa. The results showed that the macroalgal cover is generally high (≥ 50%) for low and high sea urchin densities.

Figure 8 Diadema antillarum density.

Mean (±SE) density of D. antillarum at each site surveyed (PB: Playa Baracoa and RG: Rincon de Guanabo) in the sampling periods between 2005 and 2021. The letters indicate significant changes between years.

From 2015 until 2021, a HOBO sensor recorded high temperatures in the reefs at Playa Baracoa and Rincon de Guanabo, with a mean between 28 °C and 29 °C (Fig. S1). In 2016, both reefs recorded a maximum temperature of 32 °C. The highest maximum temperature documented reached 35 °C in 2018 (Table 1).

Table 1 Sea water temperature.

Mean temperature (°C; mean ± Standard deviations (SD), minimum and maximum values) for each site surveyed (PB: Playa Baracoa and RG: Rincon de Guanabo) during the sampling period from 2012 to 2021.

Site	Year	Mean ± SD	Rank (min, max)	
	2012	28.2 ± 0.8	26.3, 29.6	
	2313	27.7 ± 1.4	20.1, 30	
	2015	28.7 ± 1.7	24.8, 31.8	
	2016	28.1 ± 1.7	23.9, 32.1	
PB	2017	27.7 ± 1.4	24.4, 31.6	
	2018	27.5 ± 1.7	22.7, 33.5	
	2019	28.9 ± 1.7	25, 33.2	
	2020	27.7 ± 1.2	24.6, 31.3	
	2021	28.9 ± 1.1	25.1, 35.4	
	2015	29.1 ± 1	26.4, 31.2	
	2016	28 ± 1.6	24.4, 31.5	
	2017	28.3 ± 1.3	24.5, 31.7	
RG	2018	28.2 ± 1.8	24, 35	
	2019	27.9 ± 2.2	24.8, 34.3	
	2021	28 ± 1.5	24.7, 30.9	

Discussion

In this study, we show that there is considerable temporal variation in Acropora density, coral cover, and other reef crest metrics over a 17-year period on two reef crests, regardless of marked differences in herbivory pressure by Diadema.

Despite recent declines, the density of A. palmata on both study sites is higher than those recorded in the Caribbean region and other reef crests in Cuba. For example, A. palmata density was 0.1 col. m−2 in Florida Keys (Miller, Chiappone & Rutten LM, 2008), in Veracruz and Banco Chinchorro, Mexico, 0.3 and 0.4 col. m−2, respectively (Vega-Zepeda, Hernández-Arana & Carricart-Ganivet, 2007; Larson et al., 2014). In St. John and St. Croix, US Virgin Islands densities were 0.04 and 0.02 col. m −2, respectively (Mayor, Rogers & Hillis-Starr, 2006; Muller, Rogers & van Woesik, 2014), and in Los Roques, Venezuela, they were to 0.1 col. 100 m−2 (Croquer et al., 2016). For Cuba, according to Caballero-Aragón et al. (2020), the mean density of A. palmata is 1 col. 10 m−1, with sites showing maximum values of up to 13 col. 10 m−1. However, these authors did not include Playa Baracoa or Rincon de Guanabo reefs in their study. It is important to point out that during some years, densities of A. palmata in our study reefs were almost double that reported by Caballero-Aragón et al. (2020).

The density of A. palmata in Playa Baracoa decreased twice during the study period, probably due to hurricane damage (Table 2, Fig. 9). The first decrease occurred between 2008 and 2012 coinciding with hurricanes Gustav and Ike and tropical storms Dolly and Paula. The second decrease occurred between 2017 and 2021. During this period there was also a decrease in A. palmata density in Rincon de Guanabo when tropical storms Irma, Michael and Alberto passed near the northwest region. Hurricanes and storms have always been integral to reef dynamics (Connell, 1978). However, the increase in frequency and intensity during recent decades (Bender et al., 2010) have reduced the resilience of this ecosystem (Gardner et al., 2005; Cheal et al., 2017). Tropical storms and hurricanes have increased in frequency in Cuba between 1980 and 2019 (Coll-Hidalgo & Pérez-Alarcón, 2021). The northwest region of Cuba is not exempt from these natural phenomena (http://www.insmet.cu, Table 2). Unfortunately, we do not have information related to the hurricanes that may have directly impacted either of these two reef crests. In addition, the heavy rains and sediment discharge associated with hurricanes may reach the coral colonies and result in partial or complete mortality (Hughes & Conell, 1999). However, it should be mentioned that hurricanes have not been the principal cause of populations of A. palmata depletion, but the joint action by several local stressors (Duran et al., 2018; Caballero-Aragón et al., 2020; Rey-Villiers et al., 2020; Rey-Villiers, Sánchez & González-Díaz, 2021).

Table 2 List of impacts on Cuban reefs.

Hurricanes and tropical storms that have affected the northwest region of Cuba just prior to or during the sampling period in this study (http://www.insmet.cu).

Year	Hurricane	
2004	Hurricane Charley (category 3)	
2005	Hurricane Dennis (category 1)	
2006	Hurricanes Rita and Wilma (category 3)
Tropical storm Alberto	
2007	Tropical storm Barry	
2008	Hurricanes Gustav and Ike (passed in the vicinity of the northwest region of Cuba)
Tropical storm Dolly	
2010	Tropical storm Paula	
2017	Tropical storm Irma	
2018	Hurricane Michael
Tropical storm Alberto	

Figure 9 Impacts on Cuban reefs.

Time series of impacts on reefs in northwestern region of Cuba during the sampling periods between 2005 and 2021.

The tendency in A. palmata mean size was to decrease in both crests, with small colonies (between 10 and 50 cm diameter) predominating, which could indicate high fragmentation caused by the impact of hurricanes (Fong & Lirman, 1995). However, in Playa Baracoa the cause of the decrease in size from 2012 to 2013 is not clear; during this time there were no hurricanes and old mortality did not increase. The small sizes could indicate that colonies are using their energy to survive in unfavorable conditions, such as those generated by pollution, and not to grow (Renegar & Riegl, 2005). The abundance of recruits (≤ five cm) increased by 2021 in the Rincon de Guanabo reef crest and was greater than in Playa Baracoa. It is impossible to determine whether these recruits resulted from larvae that settled and reached this size or are fragments that survived the hurricane(s) and successfully attached to the substrate. The high values of encrusting red algal cover recorded in 2021 in comparison to other morphofunctional groups, could facilitate larval settlement. Either way, the presence of recruits is a good indicator of recovery and resilience in this reef.

The density of A. palmata colonies in Playa Baracoa declined faster than those at Rincon de Guanabo, probably because the Playa Baracoa crest is approximately 230 m from shore and has a more significant anthropogenic influence due to its proximity to a fishing village. Water quality parameters that have been determined are the concentrations of total coliform, fecal coliform, fecal streptococcal bacteria, and δ15N, and have shown that the reefs closest to Playa Baracoa are more contaminated than those near Rincon de Guanabo (Duran et al., 2018; Rey-Villiers et al., 2020; Rey-Villiers, Sánchez & González-Díaz, 2021). Therefore, the decline in A. palmata population in Playa Baracoa may be affected by terrestrially-derived runoff discharging nutrients, pollutants, and pathogens via waste from the Latin American School of Medicine to the Santa Ana River towards the reef. By contrast, the reef crest at Rincon de Guanabo is 800 m from shore, which may reduce the nutrient influx from land runoff reaching the reef. The reef crest is preceded by an extension of 11.9 ha of mangroves (Roig-Villariño et al., 2016) and 5.9 ha of seagrasses (Aguilera, 2017). According to González-Díaz et al. (2018), the ecological connectivity between the mangrove ecosystem, seagrasses, and coral reefs makes reefs more resilient to disturbances. This connectivity may influence the better condition of the Rincon de Guanabo reef crest. However, the reef at Rincon de Guanabo could be affected by pollutants produced by oil drilling and extraction from the Boca de Jaruco thermoelectric power station. In addition, previous studies have found a water quality gradient in the northwestern region of Cuba, generated by discharge from Havana Bay and the Almendares, Cojímar and Quibú Rivers, where it has been shown that the water quality improves as it moves away from these pollution centers (Rey-Villiers et al., 2020; Rey-Villiers, Sánchez & González-Díaz, 2021).

Most A. palmata colonies (∼70%) showed signs of good health. The percentage of diseases such as white band and white pox was low in contrast to A. palmata colonies in the Caribbean, where the prevalence of white pox was 71.4% in the Florida Keys and the prevalence of white pox and white band in Akumal, Mexico, was 6% and 4%, respectively (Sutherland et al., 2016; Randazzo-Eisemann, Garza-Pérez & Figueroa-Zavala, 2022). In Playa Baracoa, old mortality predominated. It decreased in 2015, probably due to the increased re-sheeting of skeletons of A. palmata in 2013 and 2015. This form of growth in A. palmata colonies is a possible response to suitable substrate, that allows rapid regrowth of coral in a short time period, and may enhance their survival (Jordan-Dahlgren, 1992). Our data suggest that re-sheeting may be an important, but underrated, recovery mechanism for A. palmata. Rather than recruit to the substrate, where competition with algae or Millepora complanata is high, some larvae may colonize and regrow over dead A. palmata colonies. Alternatively, after episodes of partial mortality (e.g., due to disease), the healthy tissue can grow horizontally to cover recently dead skeleton by resheeting (Jordan-Dahlgren, 1992).

In both reefs, the prevalence of bleaching was highest in 2016, and, as a consequence, recent mortality increased in the 2017 survey, indicating that after thermal stress (Fig. 9), colonies were vulnerable and died. Peak bleaching registered in our study coincided with the third global bleaching event in 2014–2017, which resulted in high coral mortality on many reefs and rapid deterioration of reef structures (Eakin, Sweatman & Brainard, 2019). In 2016, the reefs at Playa Baracoa and Rincon de Guanabo recorded a maximum temperature of 32 °C (Table 1), coinciding with our peak of bleaching. After 2016, the prevalence of bleached colonies did not increase, possibly because these populations of A. palmata are more resistant to heat stress.

In addition to coral decline, algal cover increased in both reef crests. Despite the high density of herbivores in Playa Baracoa, algal abundance was not affected by D. antillarum. Algal abundance is determined by top-down control of herbivores and bottom-up control of nutrients (Smith, Hunter & Smith, 2010). In the case of Playa Baracoa, we recorded a high density of sea urchins similar to densities reported in Panama reefs before the collapse of their populations (Lessios et al., 1984). However, macroalgae cover increased during the study, indicating a strong influence of nutrients (Rey-Villiers et al., 2020; Adam et al., 2021). We recorded differences in the abundance of macroalgal groups in the two reef crests. The benthic substrate in Playa Baracoa in 2021 was characterized by lower encrusting red algal and Dictyota cover than in Rincon de Guanabo. These results may be associated with the preference of D. antillarum for certain groups of macroalgae (Maciá, Robinson & Nalevanko, 2007; Williams, 2022). Turf algal cover was low at both sites. Although, in Playa Baracoa, the values of turf algae were higher with respect to Rincon de Guanabo. Williams (2022) state that sites under urchin grazing pressure have increased turf algal cover. In this respect, it should be necessary to consider a sampling or experimental design that tests the effect of D. antillarum over algal cover, in general, or to determine if Diadema differentially influences macroalgal groups in Playa Baracoa.

Conclusions

Our results show an overall decline of A. palmata (density) and percent coral cover and a significant increase of macroalgae over 17 years in two reef crests with distinct densities of sea urchins. The coral decline seems to be driven by factors associated with anthropogenic activities and hurricanes. The disproportionate increase in macroalgal cover on both reefs between 2015 and 2021 might be due to terrestrially-derived nutrient and pollutant discharge. However, we cannot conclude that the increase in macroalgae was detrimental to the reefs until we undertake an evaluation of the variation in morphofunctional groups over time. Despite this, the two reefs studied here appear to be in better condition, with respect to A. palmata densities and health, as well as D. antillarum abundance (at Playa Baracoa only), relative to most reefs in the Caribbean. A thorough understanding of reef crest ecological drivers is crucial to develop conservation strategies and requires further investigation of other reef crests in Cuba and the Caribbean to determine whether they show similar trends and influence of ecological drivers.

Based on the results of this evaluation over 17 years, we propose a management plan whose objective is to reverse the downward trajectory of Acropora palmata populations on these two reef crests. The management plan has three components: (1) management of water quality through the reduction of local sources of pollution and nutrient inputs, (2) management of fisheries, and (3) implementation of a restoration program to outplant new coral colonies to both reef crests and to introduce D. antillarum. This management plan could improve the resilience in both reefs, by indirectly decreasing the abundance of macroalgae through improved water quality, preventing overfishing, and introducing sea urchins. Preventing the entry of pollutants into the reef could also contribute to improved coral colony health. Finally, outplanting coral fragments could improve coral cover and diversity on the reef crest.

Supplemental Information

Supplemental Information 1 Sea water temperature over time

Sea water temperature at Playa Baracoa (A) and Rincon de Guanabo (B) from 2012 to 2021. The horizontal black line inside the box represents the mean, the size of the box is the interquartile range, and the whiskers represent the minimum and maximum data values. Black circles indicate outliers.

Click here for additional data file.

Supplemental Information 2 Sample size, variables and sites surveyed

Sample size by variables across years and site. The band transect (10 m2) was the sampling unit that was used for the determinations of Acropora palmata and Diadema antillarum density. The linear transect (10 m) was employed when determining colony diameter and health as well as benthic cover.

Click here for additional data file.

Supplemental Information 3 Analyses performed on data

Analyses and R models used and the key results that were found.

Click here for additional data file.

Supplemental Information 4 Raw data

Each data point from every survey conducted from 2005 to 2021.

Click here for additional data file.

Additional Information and Declarations

Competing Interests

Author Contributions

Data Availability

Anastazia Banaszak is an Academic Editor for PeerJ.

Amanda Ramos conceived and designed the experiments, performed the experiments, analyzed the data, prepared figures and/or tables, authored or reviewed drafts of the article, and approved the final draft.

Patricia González-Díaz conceived and designed the experiments, performed the experiments, analyzed the data, authored or reviewed drafts of the article, and approved the final draft.

Anastazia T. Banaszak conceived and designed the experiments, analyzed the data, authored or reviewed drafts of the article, and approved the final draft.

Orlando Perera performed the experiments, authored or reviewed drafts of the article, and approved the final draft.

Fredy Hernandez Delgado performed the experiments, authored or reviewed drafts of the article, and approved the final draft.

Sandra Delfín de León performed the experiments, authored or reviewed drafts of the article, and approved the final draft.

Patricia Vicente Castro performed the experiments, authored or reviewed drafts of the article, and approved the final draft.

Gabriela Caridad Aguilera Pérez performed the experiments, authored or reviewed drafts of the article, and approved the final draft.

Alain Duran conceived and designed the experiments, analyzed the data, authored or reviewed drafts of the article, and approved the final draft.

The following information was supplied regarding data availability:

The raw data are available in a Supplementary File.

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
