# Peer review of "Seventeen-year study reveals fluctuations in key ecological indicators on two reef crests in Cuba"

_PeerJ, doi:10.7717/peerj.16705_

## Round 0.1 · original submission · Major Revisions

Dear Authors, three referees read the manuscript and they have given very useful and important suggestions to improve the work. They all agree that this long-time series of data on the structure of an important coral in two locations in northern Cuba is very interesting and needs to be published. Nevertheless, despite two reviewers agreeing that the paper is well written, I think that there are some small grammatical and verbosity issues to fix. Based on their revision in my opinion, the manuscript needs a major revision.

·

Basic reporting

see my comments below

Experimental design

see my comments below

Validity of the findings

see my comments below

Additional comments

This is a useful paper that described a relatively long time series of data on the structure of an important coral on two locations in northern Cuba. There are few datasets of this sort and the results are interesting. I would like to see this published.

I have a few comments, that are pretty minor to clarify methods and improve the presentation of results.

1) I see why the authors' provide densities per 10 m2 but the convention is per m2 and it'd be good to convert back to this throughout (particularly for the urchin data).

2) I'm confused by some of the results at RG. The methods state that surveys were carried out at both sites during each of the nominated years. Yet there appear to be some years where no palmata are seen at RG. Yet there density / size has increased dramatically by the next sampling interval. This seems hard to believe unless it represents severe storms followed by recruitment and rapid growth. Please clarify whether those years for which no data are presented are genuine absences and if so explain why. If not, then please explain how you can obtain such variability.

3) I felt that the statistical approaches need greater explanation. For example, what were the random effects used for if you are using mixed models? You state you use lmer for cover data yet these are bounded by 0 and 1, which means that you should be using generalised linear models with a binomial model, or beta regression. There is a statement that non-parametrics were used but it's unclear which analyses. So I'd recommend a table be included with all the questions, models used, and the key results.

4) It is intriguing that macroalgal cover is so high on shallow reef crests. That's unusual. Usually these habitats are intensively grazed by parrotfish and surgeonfish, which keeps them relatively macroalgal free even in the absence of Diadema. Are there any data or papers that can be used to comment on levels of fish herbivory? Also I really feel you need to give more details on the types of algae involved. I was unclear, for example, why you separate Dictyota from 'fleshy algae'. What other algal genera were found in the 'fleshy algae' category? Did it differ between sites or over time? Indeed, I think it would be useful to undertake a multivariate analysis of algal assemblage composition of site/time. You might use Primer for this.

5) Lastly, it'd be helpful if the time series of impacts upon these reefs could be given in a figure (e.g., when major storms, bleaching, run off etc occurred).

Nice paper and great job. It's superbly well written also.

·

Basic reporting

This is a well-written manuscript. However, the reporting structure can benefit from some edits. It is unclear if you report on reef dynamics or Acropora palmata (Apal) dynamics. The document transits between coral cover and Apal cover. There is no clarity if Apal was the only species found or if this species is the one used as an indicator species. For instance, A. cervicornis is mentioned in the introduction as a previous species, but the results do not say this species. It is not clear if it wasn't present or just not assessed. This needs to be cleared throughout the document.

In Figure 4, try to use contrasting colors. Green and blur are challenging to distinguish.

Experimental design

The experimental design is strong. My only recommendation is to review the paragraph between lines 181 and 186. You referred to a belt or band transect of 10m x 1m, but then you mentioned that the abundance of macroalgae and coral was calculated as the proportion, i.e., the total distance covered by each category divided by transect length. Here you are describing a line intercept method.

Validity of the findings

no comment

Additional comments

The last paragraph can be stronger if you flesh out your recommendations for a management plan. Why do you recommend those three components? Give examples of how those three components can help reverse the downward trajectory of Apal populations.

Reviewer 3 ·

Basic reporting

Too verbose and could do with some tightening up of the explanations – needs to be more concise. Some of the results explanations in particular are too wordy and tend to over complicate the outcome or over interpret the data i.e. stating there is a trend when its ambiguous. Whereas there are other results that are not fully delved into. And there was some missing data (e.g. line 259 - breakdown of algae cover)

Many small grammatical errors that need addressing.

Experimental design

Great long term data set for two reefs.

Measured a range of important metrics using well established techniques.

Statistical analysis seems appropriate and adequately described.

Validity of the findings

These types of long-term and detailed data sets are very important. The authors have a great temporal data set and have included a number of important metrics. However, the study is far from novel and aside from stating that these data indicate that these two reefs are doing better than other Caribbean reefs and therefore should be the focus of management initiatives, there is limited new insights. I don’t see this a reason not to publish, but given this the manuscript needs to be much improved.

Importantly, the data analysis and interpretation is undercooked – so much more could be gleaned from the data and the discussion needs to be much improved following a more thorough assessment of the data.

Additional comments

Line 105 – 106 – last part of the sentence doesn’t make sense – aren’t you already talking about ‘the rest’ of the reef crests in the region?
Line 111 to 112 – aren’t the pollutants entering coastal regions via rivers – so pollutants are the impact and rivers are the transport mechanism – so not really an ‘as well as’
Line 121: this needs more e.g. ‘the community structure at the crests of Playa Baracoa and Rincon de Guanabo have significantly changed and if they have changed to the same extent and direction over time.’
Line 138: so the MPA does not protect it from fishing?
Figure 1: How far apart are the two reefs? From the figure it looks less than 100 km. So I am just wandering if the impacts from the school pollution could also be affecting Rincon? Particularly as the main current is to the east. According to the figure - you have 6 study sites per location but no mention of these in your methods. How were these sites selected?? Randomly etc.
Line 149 to 150: Not quite sure the need for the sentence here regarding herbivores. Is this to justify why they weren’t measured? Too low in abundance?
Line 154: is this just as a count e.g. number of colonies resheeting?
Line 169: do you mean they were correlated?
Line 183: I assume this is just under the transect line and doesn’t refer to the whole 10m2 area – so line transect.
Line 212 to 218: is there some data to look at for this. A figure would be good to verify the interpretation of the data. Also – I’m unclear as to the value of recording re-sheeting? I think this needs to be explained earlier on. I would potentially take this out and focus on the more important parameters.
Figure 3: to say that the diameter had declined for RG is a bit of a stretch given that you only have 3 time points and its up and down.
Figure 4: there seems to be 3 different colours but only 2 mentioned in the caption. Also – I thought based on your methods that you were using much larger bin sizes that depicted here. This is better here but needs to better described in the methods.
Line 224: cant you do a similar figure for the 2005 data as well to show how its moved towards the left?
Line 226 to 233: could be simplified. To me the main observations are - PB – only really change in the 10 to 50 and >50, with an increase in the first and decline in the last. No real change in recruits and juveniles aside from a drop in recruits in 2006 followed by recovery. Conversely, no change in all size classes at RG until 2021 – drop in adults and a rise in recruits.
Figure 6: use the same y axis and x axis scales for both. I would also change the affectations to incidences. And I’m not sure of the value of including old and new mortality here? Or bioerosion. Mortality should be picked up in the %age cover estimates and unless you are properly assessing bioerosion rates – these are not likely to be very accurate or representative for a number of reasons. If the data here is simplified, I think the key message would be a lot clearer. In fact, I would just focus on the bleaching data (I take it there was no coral disease?) and put it in one graph – so two lines one for each site. Much better to compare temporal trends between sites.
Line 259: no mention in methods that there was going to a breakdown of algal categories. Great that you did – but again needs to be better described in the methods.
Figure 7: Coral and algal cover are really important metrics to track over time – why have you only shown 2015 and 2021?
Line 259 to 265: again where is the data – figure?? I cant see it in the supp including the raw data file?
Line 272: unclear what test you did here? Regression??
Line 266 to 274: This section could be written much more concisely e.g. The density of diadema’s were significantly greater at PB (mean value) than at RG (mean values; stats). Diadema densties were consistently low at RG over the study period (range) whereas there was a drop from X to X at PB in 2008 following which urchin densites remained at between x and x.
Question: where is the temperature data you collected? You mention it in the methods but not in the results?
Line 277: the topic sentence doesn’t match the main contents of the paragraph. Here you have focused on the point that the A.palmata densities are much higher than recorded elsewhere. So this should be your topic sentence.
Line 278: are you just referring to your diadema data here?
Line 290 to 301: you need to better link the temporal changes with the hurricane data. When did the numbers dropped – did this occur in years with hurricanes – etc. Not very convincing as written.
Line 302: OK – but where is the evidence that directly links the timeframes of declines with potential causes?
Line 305: if it’s a relative density then it could also be due to influx of recruits too. You have high CCA levels which could be facilitating this. It would be good to look at how the coral and algal cover (and the various components of this) change over time in much more detail. I think you have the data but you haven’t included it.
Line 306: survive what? What are they being exposed to that is using their energy to survive.
Line 316: at present very speculative – it would have been good to get some water quality data for this.
Line 326: a gradient from where to where???
Line 327 to 330: OK great – then start with this and not leave the evidence right to the very end of the paragraph.
Line 332: Was any disease observed?
Line 337 to 338: why not include your temperature data to back this up?
Line 341 to 343: this needs to be in your results section not buried in here! And I still haven’t seen the supp data.
Line 367: This needs to be better assessed and described before you can make this statement.
Line 368: OK but its unclear as to which macro-algae has been increasing over time. If its fleshy algae yes that’s a problem, but if its encrusting and turfing algae – much less of a problem. Where is the data so it can be properly analysed?

---

## Round 0.2 · Minor Revisions

Dear Authors, the reviewer made a revision of the manuscript and he agrees that the new version is really improved. Nevertheless, he pointed out some very few issues that need to be addressed before acceptance. So, please provide a minor revision of this current version including the reviewer's suggestions.

·

Basic reporting

n/a

Experimental design

n/a

Validity of the findings

I'm happy with the revision with one exception. I feel like there's some confusion about the appropriate choice of statistical model. I see that negative binomial regression models have been used for several analyses. In each case it seems like the response data are continuous (densities, sizes). I would usually expect to see a nbinom used for highly clustered COUNT data (number of corals). So I think you need to reconsider whether that statistical approach is desirable as its based on a discrete distribution.

Typically for density and size data I'd use linear models either based on a normal distribution or possibly gamma. Feel free to use transformations is the model diagnostics aren't good.

Also be aware that when you model cover, this is limited to between 0-1 (0-100%) and therefore you should use a binomial model or beta regression - both of which can be done in the glmmTMB.

Additional comments

n/a

---

## Round 0.3 · accepted · Accept

Dear Authors, as Editor, I found the paper improved so I am happy to inform you that your paper is accepted for publication in PeerJ. Congratulations!